1       **Introducing the MISR Level 2 Near Real-Time Aerosol Product**

4       Marcin. L. Witek[1], Michael J. Garay[1], David J. Diner[1], Michael A. Bull[1], Felix C. Seidel[1], Abigail

5       M. Nastan[1], and Earl G. Hansen[1]

7       [1]Jet Propulsion Laboratory, California Institute of Technology, 4800 Oak Grove Drive,

8       Pasadena, CA 91109, USA

Corresponding author: Marcin L. Witek, email: marcin.l.witek@jpl.nasa.gov
**Abstract**
Atmospheric aerosols are an important element of Earth's climate system, and have significant
impacts on the environment and on human health. Global aerosol modeling has been
increasingly used for operational forecasting and as support to decision making. For example,
aerosol analyses and forecasts are routinely used to provide air quality information and alerts in
both civilian and military applications. The growing demand for operational aerosol forecasting
calls for additional observational data that can be assimilated into models to improve model
accuracy and predictive skill. These factors have motivated the development, testing, and
release of a new near real-time (NRT) level 2 (L2) aerosol product from the Multi-angle Imaging
SpectroRadiometer (MISR) instrument on NASA's Terra platform. The NRT product capitalizes
on the unique attributes of the MISR aerosol retrieval approach and product contents, such as
reliable aerosol optical depth as well as aerosol microphysical information. Several
modifications are described that allow for rapid product generation within a three-hour window
following acquisition of the satellite observations. Implications for the product quality and
consistency are discussed as compared to the current operational L2 MISR aerosol product.
Several ways of implementing additional use-specific retrieval screenings are also highlighted.

## 1. Introduction

Atmospheric aerosols have for long been recognized to influence the climate, environment, and human health (e.g., IPCC, 2013; Lelieveld et al., 2015; Shindell et al., 2013; Turnock et al., 2020). They also affect satellite remote sensing of important geophysical parameters such as ocean color (e.g., Frouin et al., 2019; Gordon, 1997) or greenhouse gas abundance (Butz et al., 2009; Frankenberg et al., 2012; Houweling et al., 2005). Aerosol particles and their properties have been extensively studied in-situ and remotely: from the ground, in the air, and from space. These observational data vary in spatial and temporal coverage, but usually only offer snapshots of local conditions. Since atmospheric aerosols have a life cycle ranging from hours to days, numerical modeling of their emission, transport, and deposition has filled the coverage gaps and extended our understanding of their global impacts. This has given rise to a number of global aerosol reanalyses (Buchard et al., 2017; Gelaro et al., 2017; Inness et al., 2013, 2019; Lynch et al., 2016; Randles et al., 2017; Rienecker et al., 2011) that provide a long-range, gridded, and internally consistent outlook on aerosol burdens around the world. Furthermore, global aerosol modeling has been increasingly used for operational forecasting (e.g., Xian et al., 2019) and as support to decision making, for example in air quality alerts and in non-civilian applications (Liu et al., 2007).

The growing demand for consistent gridded aerosol products has been driving development and steady improvement of numerical predictions. For example, the International Cooperation for Aerosol Prediction initiative was founded in 2010 (Benedetti et al., 2011; Reid et al., 2011), with one of its goals being the development of global multi-model aerosol forecasting ensemble for basic research and operational use (Xian et al., 2019). Still, models suffer from often poorly resolved aerosol emissions and sinks and can be affected by errors in the underlying meteorology. As a result, systematic and sampling-related biases in aerosol fields are often found between model simulations and satellite observations (e.g., Buchard et al., 2015; Colarco et al., 2010; Lamarque et al., 2013; Zhang and Reid, 2009). An effective way to mitigate some of these problems is by assimilating aerosol observations into numerical models (e.g., Bocquet et al., 2015; Fu et al., 2017; Sekiyama et al., 2010; Di Tomaso et al., 2017; Werner et al., 2019; Zhang et al., 2008). Satellite observations of aerosol optical and microphysical properties are inseparable from these data assimilation activities as they offer the necessary data volume, near-global coverage, and frequent repeat cycle. However, an often-considerable latency for generating science-quality "standard" satellite products (8 to 40 hours) renders them unsuitable for operational forecasting. This has led to the development of aerosol

products within the time frame required by modeling centers, usually three hours from satellite
overpass. A number of near real-time (NRT) products has emerged.

One example of a platform that provides users with NRT satellite products and imagery

is NASA's Land, Atmosphere Near real-time Capability for EOS (LANCE) project
(https://earthdata.nasa.gov/earth-observation-data/near-real-time). A range of instruments
deliver various Level 1 (L1) and Level 2 (L2) data products
(https://earthdata.nasa.gov/collaborate/open-data-services-and-software/data-information-
policy/data-levels), including radiances, land surface properties, and atmospheric
thermodynamics and composition within three hours from satellite observation. NRT aerosol
products are currently available from the Moderate Resolution Imaging Spectroradiometer
(MODIS), Ozone Monitoring Instrument (OMI), and Visible Infrared Imaging Radiometer Suite
(VIIRS). NASA's Multi-angle Imaging SpectroRadiometer (MISR) currently provides NRT
radiance and cloud motion vector products. The purpose of this paper is to introduce a new
MISR NRT L2 aerosol product available within LANCE.

This paper is organized as follows. Section 2 and 3 provide brief descriptions of the

MISR instrument and the data processing sequence, respectively. Section 4 first outlines the
cloud identification methods employed in the MISR aerosol algorithm and then describes
algorithmic modifications introduced in the NRT processing. Adjustments to cloud and retrieval
screening parameters and their implications are discussed. The global distributions of the NRT
product and comparisons of total and fractional AODs with the standard aerosol product are
presented in Section 5. Section 6 provides a summary.

**2. MISR instrument and aerosol data product**

The MISR instrument flies aboard the NASA Earth Observing System (EOS) Terra satellite,
launched in December 1999 to a sun-synchronous descending polar orbit, at an orbital altitude
of 705 km, an orbital period of 99 minutes, and an equatorial crossing time of 10:30 a.m. local
time. MISR makes 14.56 orbits per day with a repetition cycle (revisit) of 16 days. The orbit
tracks are georeferenced to a fixed set of 233 ground paths. With a cross-track swath of about
380 km, total Earth coverage is obtained every 9 days at the equator and every 2 days at high
latitudes.

MISR contains nine pushbroom cameras with viewing angles at the Earth's surface

ranging from 0° (nadir) to +/- 70.5° oriented along the direction of the flight track. A point on the
ground is imaged by all nine cameras in approximately 7 minutes. The cameras make
observations of reflected solar radiance in four spectral bands, centered at 446 (blue), 558
(green), 672 (red), and 866 (near-infrared) nm. The spatial resolution depends on the camera
and wavelength. The red band has a full 275 m resolution in all cameras. The other three
spectral channels are averaged onboard to a 1.1 km resolution in global-mode operation (Diner
et al., 1998), with the exception of the nadir camera which preserves the full 275 m resolution in
all spectral channels. See https://misr.jpl.nasa.gov/Mission/ for more details.

MISR employs two processing pathways for aerosol retrievals, one for observations over

land (Martonchik et al., 2009), and another for dark water (DW) (Kalashnikova et al., 2013),
which applies over deep oceans, seas, and lakes. Previous versions of the MISR aerosol
product were extensively validated over the years (e.g., Kahn et al., 2010; Kahn and Gaitley,
2015; Kalashnikova et al., 2013; Shi et al., 2014; Witek et al., 2013) showing high retrieval
quality over land and ocean.

The current operational version of the MISR aerosol product, designated as version 23

(V23), was released publicly in June 2018. It introduced multiple algorithmic, data product, and
data usability improvements (Garay et al., 2020; Witek et al., 2018a, 2018b). V23 provides
aerosol information with a spatial resolution of 4.4 km x 4.4 km packaged in NetCDF-4 format.
Initial validation efforts showed that V23 retrievals are more accurate than previous versions,
with most pronounced improvements in the DW algorithm (Garay et al., 2020). V23 retrievals
over oceans were extensively validated by Witek et al. (2019), indicating excellent agreement
with ground-based observations. Other V23 Aerosol Optical Depth (AOD) evaluation efforts
show similar results (e.g., Choi et al., 2019; Sayer et al., 2020; Si et al., 2020; Sogacheva et al.,
2020). A first regional insight into retrieved particle properties from the MISR V23 aerosol
product shows that MISR generally captures the distinct spatial and temporal features of aerosol
type in East Asia (Tao et al., 2020). Furthermore, V23 has greatly improved the quality of
reported AOD uncertainties, which now realistically represent retrieval errors (Sayer et al., 2020;
Witek et al., 2019). This is especially relevant as pixel-level retrieval uncertainties are very
important for satellite data assimilation, which is being increasingly used in aerosol modeling
studies (Lynch et al., 2016; Shi et al., 2011, 2013; Zhang and Reid, 2010). MISR data and
related documentation can be obtained from: https://asdc.larc.nasa.gov/project/MISR.

**3. NRT latency and data description**

MISR currently provides several L1 and L2 near real-time (NRT) radiance and cloud motion
vector products (https://earthdata.nasa.gov/earth-observation-data/near-real-time/download-nrt-
data/misr-nrt).  All MISR NRT processing is based on Level 0 data downlinked in observational
sessions. These session-based files, representing portions of a single MISR orbit, usually cover
between 10 to 50 minutes of observations, as compared to the full orbit period of 98.9 minutes.
This session-based processing is necessary to allow for the fast product delivery required for
NRT applications.

The new NRT L2 aerosol product file content, described in Data Product Specification

(https://asdc.larc.nasa.gov/documents/misr/DPS_AEROSOL_NRT_V023.20210430.pdf), is
equivalent to the standard aerosol product (Garay et al., 2020). The NRT L2 aerosol product file
name convention is:
MISR_AM1_AS_AEROSOL_T{yyyymmddHHMMSS}_P{ppp}_O{oooooo}_F13_0023.nc, where
'yyyy', 'mm', and 'dd' are the year, month, and day, and 'HH', 'MM' and 'SS' are the hour,
minute, and seconds, respectively. Furthermore, {ppp} is the three-digit path identifier (between
001 and 233) and {oooooo} is the six-digit orbit number. The NRT L2 aerosol product files are
available for download within three hours of acquisition at NASA's Atmospheric Science Data
Center (ASDC) (https://asdc.larc.nasa.gov/project/MISR).

For clarity, it is important to distinguish between the three different MISR L2 aerosol

products: NRT, FIRSTLOOK, and standard aerosol (SA) product (see Figure 1). NRT is
generated within a three-hour time interval after acquisition and uses the same ancillary inputs
as FIRSTLOOK. These include the monthly gridded (1.0 degree) snow/ice mask and surface
wind speed from the Terrestrial Atmospheric and Surface Climatology (TASC) database and the
seasonal Radiometric Camera-by-camera Threshold Dataset (RCTD) (Diner et al., 1999a). Both
NRT and FIRSTLOOK utilize TASC and RCTD datasets from the current month/season in the
prior year. The FIRSTLOOK product is generated within two days from acquisition and includes
cloud classification parameters obtained from the L1 and L2 cloud products. The SA product is
available after final processing is performed on a seasonal basis and within three months past
the end of the season, which results in a 3–6-month latency. The final processing utilizes the
most recent snow/ice and wind speed data.

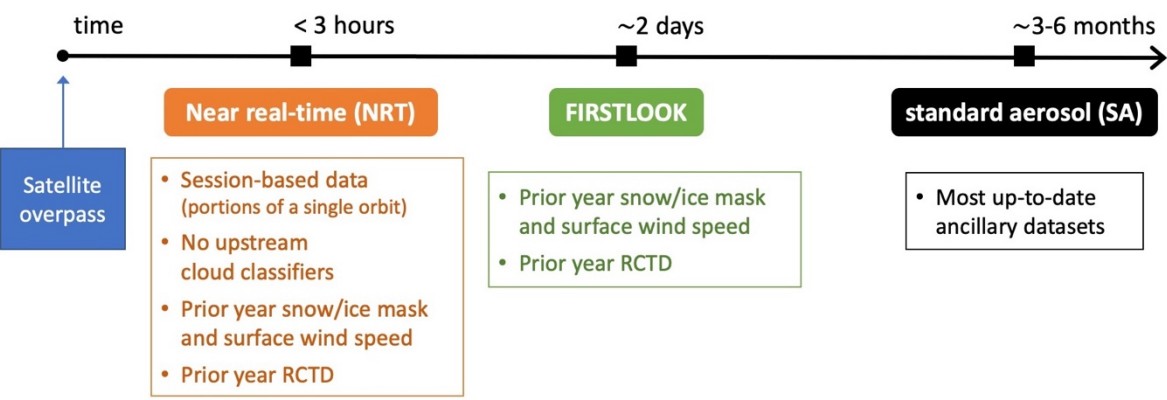

Figure 1 *Schematic showing MISR aerosol product delivery timeline. Snow/ice mask and surface wind speed data are monthly*
*averages. RCTD stands for Radiometric Camera-by-camera Threshold Dataset. MISR final production (SA) is processed on a*
*seasonal cycle and is often delayed one to three months past the end of each season, which results in up to 6-month latency.*

## 165 4. Cloud screening in the NRT MISR aerosol product


### 167 4.1. Cloud identification


Identification of cloudy pixels is a critical element of all satellite aerosol remote sensing
algorithms. MISR employs several cloud identification strategies which can be loosely split into
two groups: the first group relies on cloud classifiers previously generated with MISR Level 2
Cloud Detection and Classification algorithm (Diner et al., 1999b), and the second group
includes build-in tests that are internal to the aerosol retrieval algorithm (Diner et al., 2008).

### 175 4.1.1. Upstream cloud classifiers


The operational MISR aerosol algorithm relies on a range of external input datasets that are
either static—for example, a monthly wind speed climatology—or that need to be generated
prior to aerosol retrievals in upstream processing. A notable example of such external inputs to
the SA and FIRSTLOOK algorithms are cloud classification parameters obtained from the MISR
L2 cloud product. An important implication of this dependency is that aerosol processing needs
to wait for the cloud product to be generated, creating a time lag that is prohibitive for NRT
applications. Typically, the L2 cloud product is generated within about 18 hours of overpass,
and the MISR L2 FIRSTLOOK aerosol processing is completed within about 2 days. In order to
produce an L2 aerosol product within an about three-hour time frame, the algorithm needs to
operate without the upstream cloud classifiers.
Two specific L2 cloud classification parameters utilized in FIRSTLOOK and SA aerosol
processing are the MISR Stereoscopically-Derived Cloud Mask (SDCM) and the Angular
Signature Cloud Mask (ASCM) (Diner et al., 1999b; Girolamo and Davies, 1994). In addition to
these L2 products, the Radiometric Camera-by-camera Cloud Mask (RCCM) (Diner et al.,
1999a; Girolamo and Davies, 1995) retrieved in L1B processing is also employed. All three
parameters are reported at 1.1 km x 1.1 km resolution. It should be noted that RCCM also
serves as an input to the algorithm that generates SDCM and ASCM, indicating that these
parameters are not independent.
In the FIRSTLOOK and SA algorithm, the RCCM, SDCM, and ASCM cloud masks are
used together to determine whether a particular 1.1 km x 1.1 km subregion is clear or cloudy.
The implication is that if any of the 9 MISR cameras is designated as cloudy in a subregion, this
subregion is excluded from aerosol retrieval. The clear/cloudy decision logic depends on the
underlying surface type, assigned into three categories: land, water, and snow/ice. Generally, a
"clear" outcome is favored over the two most frequently used surface types, land and water,
assigning a subregion as cloudy only if the RCCM and SDCM masks indicate a cloud. The logic
is considerably more conservative over snow/ice surfaces due to difficulties in distinguishing
clouds from the underlying bright features. Details of the cloud mask decision logic over different
surface types can be found in Diner et al. (2008).
Analyzing three months of V23 L2 SA product (March, April, May, 2020) indicates that
the cloud masks along with the brightness test (see 4.1.2) lead to screening of about 50% of
retrievals. As such, they have the largest impact on identifying and removing pixels where
clouds might be present. These masks and decision pathways, however, have their deficiencies
and additional checks were put in place to further decrease the frequency of cloud-
contaminated aerosol retrievals.

**4.1.2. Built-in cloud detection methods**

In addition to the cloud masks retrieved in the L1B processing (RCCM) and from the L2 Cloud
Detection and Classification algorithm (SDCM, ASCM), the MISR aerosol retrieval algorithm
relies on three internal tests to further identify cloudy pixels that might have escaped earlier
detection. These are (1) the *brightness test*, (2) the *angle-to-angle smoothness test*, and (3) the
*angle-to-angle correlation test*. Details of these tests can be found in Martonchik et al. (2002) or
Diner et al. (2008), but a short summary is provided here for completeness.
The brightness test is employed to identify clouds that lacked sufficient texture to be
picked up by SDCM. For each surface type a fixed threshold is adopted on measured
bidirectional reflectance factors (BRFs), and when exceeded in all spectral bands for at least
one camera, it renders a subregion unsuitable for aerosol retrieval. The thresholds are set to
1.0, 0.5, and 0.5 for snow/ice, land, and water surfaces, respectively. The value of 1.0 means
that the brightness test is effectively turned off over snow/ice. Furthermore, the brightness test
does not override subregions that were identified as clear by RCCM.
The angular smoothness test checks for unusually large variations in the measured
equivalent reflectances as a function of camera angle, the premise being that in the absence of
artifacts or subpixel clouds, the measured radiance should change smoothly from camera to
camera. The test is achieved by fitting a polynomial to equivalent reflectances, separately for aft
(+nadir) and forward (+nadir) cameras and each spectral band, and checking if the goodness of
fit metric (definition in Diner et al., 2008) exceeds a threshold. If in at least one case the test
fails, the subregion is eliminated.
Finally, the angle-to-angle correlation test also investigates radiance smoothness and
correlation between camera angles, which makes it conceptually similar to the angular
smoothness test, but instead utilizes high-resolution information from the red spectral band. It
uses 4 x 4 arrays of the 275m spatial resolution red band equivalent reflectances in each 1.1 km
x 1.1 km subregion. The test then evaluates spatial variability within the 4 x 4 array for each
camera and compares it to a variability within a camera-average template. Variances,
covariances, and normalized cross-correlations are calculated (see Diner et al., (2008) for
details). If the variability within a camera deviates considerably from the average, this camera
might have sub-pixel clouds or other contaminants, and as a result the subregion is excluded
from aerosol retrievals.
In the three months of data analyzed in this study (March, April, May 2020), the relative
occurrence of retrieval screening due the above-mentioned internal tests are about 4.0% and
0.1% for the correlation and smoothness tests, respectively. These statistics come from
analyzing the output field *Aerosol_Retrieval_Screening_Flags* and as such they do not
represent the absolute rates of success of each individual test. That is because the tests are
performed sequentially, and if one fails, subsequent tests are not performed. For SA product
generation, the order is: upstream cloud mask described in 4.1.1, the brightness test, the
correlation test, and the smoothness test. For example, the correlation test is only performed on
pixels that already passed the upstream cloud tests as well as the brightness test. Additionally,
the brightness test does not have its own flag in the *Aerosol_Retrieval_Screening_Flags* output
but is grouped together with the upstream cloud classifiers.

**4.2. Retrieval screening using regional cloud parameters**

Methods described in section 4.1 focus on identifying and excluding cloudy 1.1 km x 1.1 km
subregions from the aerosol retrieval process. The retrieval region consists of 16 (4 x 4)
subregions. These methods are highly effective at removing cloud-contaminated pixels, but
since they rely on MISR visible wavelengths they might miss certain cloud signatures more
easily detected in the infrared spectrum (e.g., Gao et al., 1993). For example, MODIS routinely
uses its reflective and emissive infrared channels to detect optically thin cirrus clouds
(Ackerman et al., 2010; Levy et al., 2013). As a result, MISR cloud detection methods
occasionally fail, which leads to visible outliers in retrieved AODs (Witek et al., 2018b). For that
reason, an additional set of screenings is applied in an effort to eliminate such unusually high
AOD retrievals (Garay et al., 2020). Two of these additional methods look at overall cloudiness
in the retrieval region (consisting of 4 x 4 subregions) as well as in a larger area consisting of 3
x 3 regions (12 x 12 subregions). The Cloud Screening Parameter (CSP) represents the fraction
of clear grid cells within a region, whereas Cloud Screening Parameter Neighbor 3x3 (CSP9) is
similar to CSP but for the larger area. If CSP is below 0.7 and CSP9 below 0.5, the retrieval is
not reported in the final product intended for most users. However, it is still included in the
product's AUXILIARY subcategory and annotated with the term "Raw" to indicate that the
product has not passed the recommended quality screenings.

**4.3. Adjusting cloud screening thresholds**

**4.3.1. Performance of the prototype NRT product**

This subsection presents results and analysis of prototype NRT aerosol retrievals. These are
obtained prior to any threshold and screening adjustments included in the final version of the
product. To differentiate between the final and the prototype NRT products, the latter is donated
as $NRT_{prot}$.
As mentioned in the previous section, the NRT processing cannot rely on the cloud
masks generated in the L1 and L2 cloud products, namely the RCCM, SDCM, and ASCM. This
implies that potentially less screening of cloudy subregions would be applied, increasing the
probability of cloud contamination in aerosol retrievals. However, some of the burden of cloud
identification is picked up by the built-in cloud tests described in section 4.1.2. The frequency of
these tests identifying cloudy pixels increases in NRT processing in comparison to standard
processing, in large part mitigating the negative consequences resulting from the lack of the
upstream cloud masks. This is well evidenced by examining the normalized probability density
functions (*pdf*s) of AOD from spring 2020 (Figure 2). The SA (red) and $NRT_{prot}$ (blue) lines are
very similar, indicating that the built-in cloud tests substitute to a significant extent for the
missing upstream cloud masks in generating the $NRT_{prot}$ product. The largest difference occurs
in the high-AOD range, suggesting that $NRT_{prot}$ has more retrievals in this regime. The black
dotted line shows a *pdf* of the $NRT_{prot}$ AOD retrievals that do not have a matching SA retrieval.
This is labeled as "$NRT_{prot}$ gained" as it represents additional retrievals obtained in NRT
processing due to the lack of external cloud masks. The "$NRT_{prot}$ gained" *pdf* is clearly shifted
towards higher AODs, confirming that the $NRT_{prot}$ processing tends to retrieve higher AODs in
places where SA is not available.

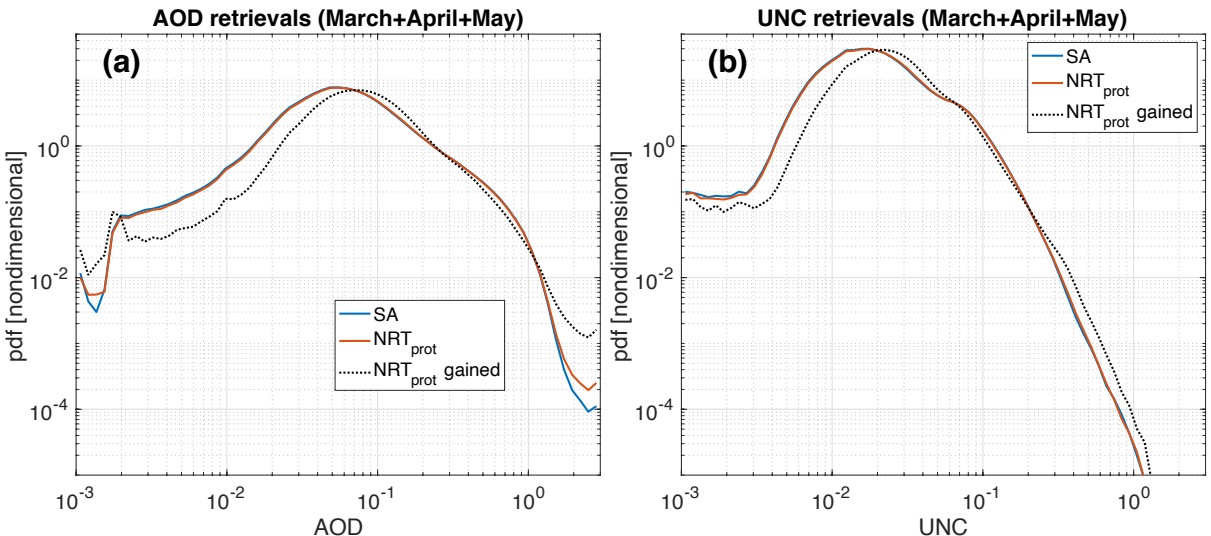


*Figure 2 (a) AOD normalized probability density functions from SA, prototype NRT, and prototype NRT retrievals that do not*
*have a matching SA equivalent (labeled as $NRT_{prot}$ gained); (b) same as in (a) but for retrieved AOD uncertainties (UNC). Data*
*statistics for AODs are provided in Table 1.*

Figure 3 shows *pdf*s of AOD but with retrievals separated between DW (Fig. 3a) and

land (Fig. 3b). These *pdf*s indicate that the retrievals over oceans are the main source of
increased frequency of high-AODs in the $NRT_{prot}$ product. The *pdf*s over land are virtually
unchanged, including a slightly flattened but still relatively comparable distribution of the "$NRT_{prot}$
gained" retrievals (Fig. 3b). The additional statistics of the data presented in Figs. 2 and 3,
including the retrieval count, the mean AOD, and the geometric mean AOD, which is better
suited for log-normal distributions of AOD (Sayer and Knobelspiesse, 2019), are provided in
Table 1. Note that the number of $NRT_{prot}$ gained is not the same as the number of $NRT_{prot}$ minus
SA. This is because some SA retrievals do not have their $NRT_{prot}$ equivalent, making the SA
count larger than it would have been otherwise.

In the 3-month period analyzed in this study (March, April, May, 2020), the $NRT_{prot}$

processing leads to about 6.4% more retrievals than SA (see Table 1). 5.5 million $NRT_{prot}$
retrievals do not have a matching SA retrieval (NRT gained), and the majority of them (67%) are
DW retrievals. The overall geometric means are almost identical in SA and $NRT_{prot}$, although
small variations in this statistic are seen in DW and land categories. The NRT gained have
visibly higher arithmetic and geometric mean values, the increase coming mainly from DW
retrievals. These basic statistics warrant a further look at the $NRT_{prot}$ performance over DW.

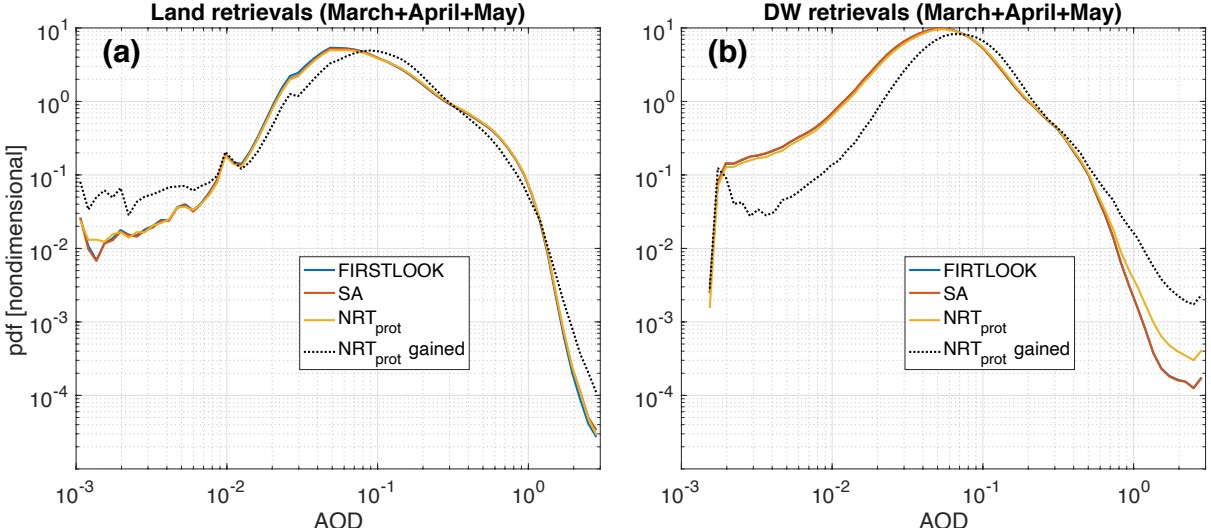


*Figure 3 AOD pdfs for land (a) and DW (b) retrievals, respectively. Data statistics are provided in Table 1.*

|  | All retrievals | | | DW | | | Land | | |
|---|---|---|---|---|---|---|---|---|---|
|  | SA | $NRT_{prot}$ | $NRT_{prot}$ gained | SA | $NRT_{prot}$ | $NRT_{prot}$ gained | SA | $NRT_{prot}$ | $NRT_{prot}$ gained |
| $N$ ($\times 10^6$) | 49.7 | 52.9 | 5.5 | 27.6 | 30.7 | 3.7 | 22.1 | 22.2 | 1.8 |
| *mean* | 0.168 | 0.169 | 0.171 | 0.111 | 0.115 | 0.146 | 0.240 | 0.243 | 0.224 |

| | | | | | | | | | |
|---|---|---|---|---|---|---|---|---|---|
| *geomean* | 0.111 | 0.112 | 0.122 | 0.083 | 0.085 | 0.106 | 0.160 | 0.162 | 0.161 |

*Table 1 Additional statistics for the data presented in Figs. 2 and 3 (statistic for FIRSTLOOK not shown). NRT gained stands for*

*the prototype NRT retrievals that do not have a matching SA equivalent; geomean stands for the geometric mean AOD.*

### 4.3.2. Sensitivity to CSP and CSP9 thresholds in DW retrievals

One way to screen potentially cloud-contaminated high-AOD retrievals is to adjust thresholds on CSP and CSP9 parameters (Garay et al., 2020). This is furthermore justified by the fact that in the absence of RCCM, SDCM, and ASCM in $NRT_{prot}$ processing, fewer cloudy subregions are identified in a retrieval area and consequently CSP and CSP9 have by default lower values. This argument provides strong justification for investigating sensitivity to increased CSP and CSP9 thresholds in the $NRT_{prot}$ processing.

The SA product uses the thresholds of CSP=0.7 and CSP9=0.5 (Garay et al., 2020); when the values of CSP and CSP9 are below these thresholds in a retrieval region, the aerosol retrieval is removed from the data field recommended for users. Figure 4 and Table 2 show *pdf*s and AOD statistics for different thresholds of CSP and CSP9 parameters in the $NRT_{prot}$ product over dark water surfaces. There are only minor changes in the *pdf*s when the thresholds are increased, including in the high-AOD regime. The arithmetic and geometric mean values decrease slowly; even at the highest considered thresholds (0.85 for CSP and 0.75 for CSP9) these statistics are still above the SA values. At the same time the number of passing $NRT_{prot}$ retrievals decreases considerably faster, with almost 19% of retrievals lost when the highest thresholds are used. These results indicate that adjusting CSP and CSP9 thresholds is not an effective strategy to constraining $NRT_{prot}$ retrievals.

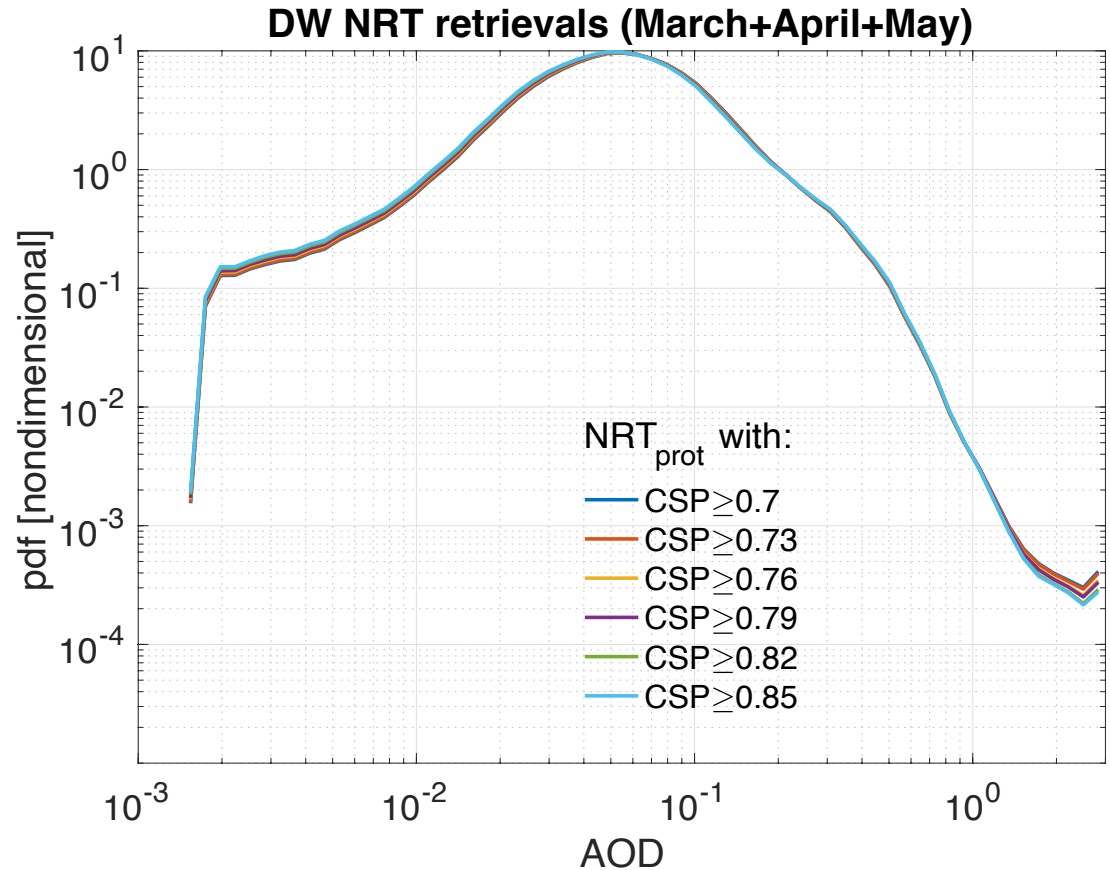


*Figure 4 Prototype NRT AOD pdfs over dark water surfaces from spring 2020 obtained with different CSP and CSP9 cloud-*
*screening thresholds. Data statistics are provided in Table 2.*

| $N$ (×$10^6$) | 30.7 | 30.1 (-1.9%) | 28.4 (-7.4%) | 27.7 (-9.8%) | 25.9 (-15.6%) | 24.9 (-18.9%) | SA 27.6 |
|---|---|---|---|---|---|---|---|
| *CSP* | ≥0.7 | ≥0.73 | ≥0.76 | ≥0.79 | ≥0.82 | ≥0.85 | |
| *CSP9* | ≥0.5 | ≥0.55 | ≥0.6 | ≥0.65 | ≥0.7 | ≥0.75 | |
| *mean* | 0.1151 ± 0.1200 | 0.1149 ± 0.1199 | 0.1145 ± 0.1190 | 0.1144 ± 0.1191 | 0.1142 ± 0.1185 | 0.1143 ± 0.1189 | 0.1110 ± 0.1079 |
| *geomean* | 0.0850 | 0.0847 | 0.0841 | 0.0839 | 0.0834 | 0.0832 | 0.0826 |

*Table 2 Additional statistics for the data presented in Fig. 4. Values for CSP and CSP9 indicate their corresponding thresholds for*
*screening AOD retrievals. The arithmetic mean values are accompanied by their respective ± one standard deviations.*

**4.3.3. Sensitivity to ARCI threshold in DW retrievals**

V23 of the MISR aerosol product introduced a new parameter, called the aerosol retrieval
confidence index (ARCI), that is used to screen high-AOD retrieval outliers caused by cloud
contamination and other factors (Witek et al., 2018b). ARCI, defined only for DW retrievals,
proved to be an efficient metric at filtering out potentially cloud-contaminated AOD retrievals. In
standard processing, retrievals with ARCI < 0.15 are removed from the recommended user
field, but are retained in the AUXILIARY group. The 0.15 threshold is well supported through
statistical analysis (Witek et al., 2018b), although some erroneous results still pass this
screening method, suggesting that increasing this threshold might be beneficial in NRT
processing.
Figure 5 and Table 3 show *pdf*s and AOD statistics for different thresholds of ARCI in the
$NRT_{prot}$ product. In this case the differences between ARCI thresholds are quite noticeable,
especially in the high-AOD range of retrievals. Increasing the ARCI threshold to 0.2 leads to a
loss of about 11% of $NRT_{prot}$ DW retrievals, but the resulting arithmetic and geometric mean
values are lower than the SA values. At the same time, the absolute number of $NRT_{prot}$ DW
retrievals (27.4 million) is still comparable to the number of SA DW retrievals (27.6 million). The
*pdf*s and the statistics suggest that increasing the $NRT_{prot}$ ARCI threshold from 0.15 to 0.18
leads to a product that has similar characteristics to SA.

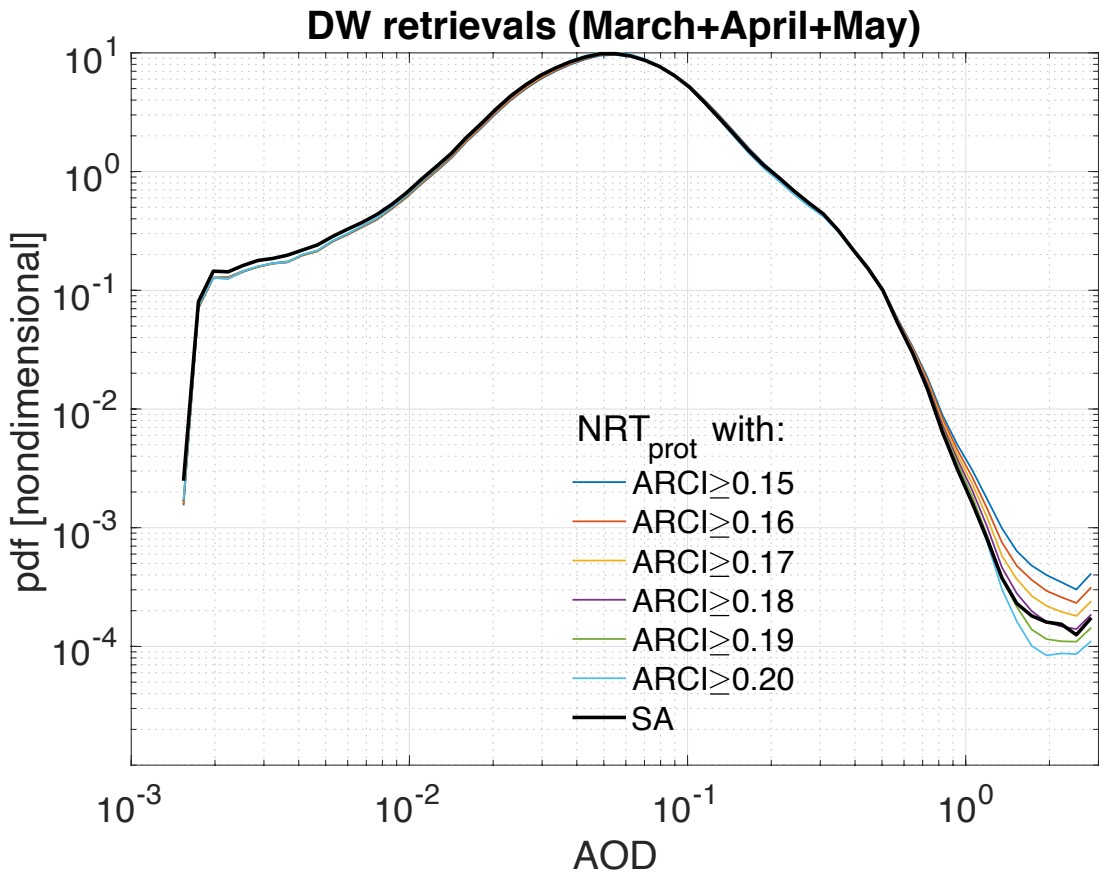


*Figure 5 Prototype NRT AOD pdfs from spring 2020 obtained with different ARCI thresholds. Data statistic are provided in Table*
*3.*

| $N$ (×10⁶) | 30.7 | 30.0 (-2.2%) | 29.4 (-4.3%) | 28.7 (-6.5%) | 28.0 (-8.6%) | 27.4 (-10.8%) | SA 27.6 |
|---|---|---|---|---|---|---|---|
| *ARCI* | ≥0.15 | ≥0.16 | ≥0.17 | ≥0.18 | ≥0.19 | ≥0.20 | |
| *mean* | 0.1151 ± 0.1200 | 0.1137 ± 0.1157 | 0.1124 ± 0.1122 | 0.1112 ± 0.1094 | 0.1100 ± 0.1070 | 0.1090 ± 0.1051 | 0.1110 ± 0.1079 |
| *geomean* | 0.0850 | 0.0842 | 0.0835 | 0.0828 | 0.0821 | 0.0813 | 0.0826 |

*Table 3 Additional statistic for the data presented in Fig. 5.*

**4.3.4. Recommendation for NRT processing**

The statistical analyses presented in the previous sections indicate that the lack of RCCM,
SDCM, and ASCM in NRT processing has negative consequences on the product, especially by
allowing more, potentially cloud-contaminated, high-AOD DW retrievals to pass screening
criteria. Adjusting build-in cloud screening thresholds on CSP and CSP9 brings only limited
benefits at the cost of losing a considerable percentage of retrievals. However, the ARCI
threshold adjustments result in much closer statistical correspondence between the $NRT_{prot}$ and
standard AOD retrievals. For that reason, a revised ARCI threshold of 0.18 is implemented in
NRT processing. Since the unscreened retrievals, as well as the ARCI parameter, are also
provided in the AUXILIARY group of the product, users are encouraged to experiment with their
own thresholds which might prove more beneficial in specific applications or geographic areas.

**4.4. Cloud/clear decision logic over snow/ice**

In section 4.1.1 the impact of upstream cloud classifiers in standard processing—namely the
RCCM, SDCM, and ASCM—on the subregion's cloud/clear designation was briefly described.
The decision pathway depends on the underlying surface type, which can be either land, water,
or snow/ice. Over land and water, the "cloud" outcome is only obtained when both RCCM and
SDCM designate the subregion as cloudy. In the absence of RCCM and SDCM the default
outcome is "clear". Over snow/ice, however, the logic is more restrictive and favors the "cloudy"
designation (Diner et al., 2008). Specifically, when the upstream cloud classifiers are not
available, the subregion designation is set to "cloudy" by default. This has important implications
on aerosol retrievals in areas where snow and ice occur seasonally.

The snow/ice surface mask, unlike land and water, is not static and changes every

month. Furthermore, the snow/ice mask input to MISR aerosol processing has a 1.0-degree
horizontal resolution, which is re-gridded to a 1.1 km resolution corresponding to the resolution
of MISR subregion. In FIRSTLOOK processing, the snow/ice mask from the same month but in
the previous year is used. The final SA processing is performed when the current year's monthly
snow/ice mask becomes available. The NRT processing, similarly to FIRSTLOOK, relies on the
previous year's snow/ice mask. Additionally, given the lack of upstream cloud classifiers, the
snow/ice areas are designated as "cloudy" for aerosol retrieval purposes. This is well visualized
in Figure 6 which shows the visible image and the corresponding maps of AOD and Aerosol
Retrieval Screening Flag in the NRT processing. The dark blue color (index 5) denotes cloudy
regions determined using the snow/ice cloud logic. The box-like nature of the excluded areas is
associated with the coarse resolution of the snow/ice mask (1.0 degree). The previous year's
mask might also not be representative of the current conditions on the ground. It is worth noting
that the FIRSTLOOK product often suffers from the same exclusion rules as NRT. This is

because of the strict clear/cloud logic over snow/ice surfaces which favors the cloudy outcome; in the case shown in Fig. 6 the AOD gaps in FIRSTLOOK (not shown) look very similar to the NRT product.

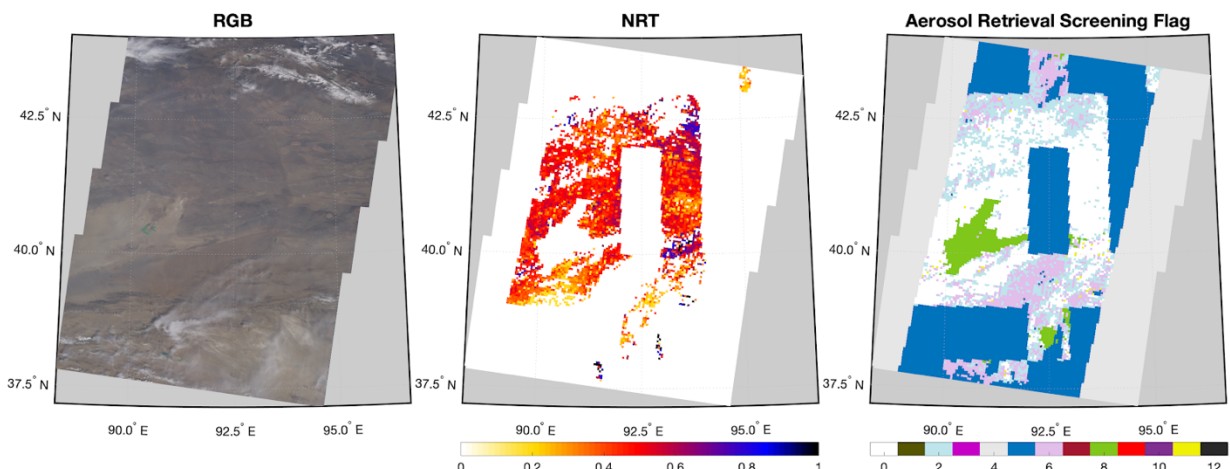

*Figure 6 Example of snow/ice masking in NRT AOD retrievals. (Left) Visible image of the retrieval area. (Center) Corresponding NRT AOD retrievals. (Right) NRT Aerosol Retrieval Screening Flag for the same area; the dark blue color denotes regions designated as cloudy.*

Several attempts have been made by the MISR science team to improve NRT aerosol retrievals in snow/ice covered areas. However, identifying and isolating snow-covered surfaces in the absence of upstream cloud classifiers proves very challenging. The quality of aerosol retrievals is often negatively affected in such conditions. For that reason, and in an attempt to eliminate as many NRT AOD outliers as possible, the current snow/ice logic is retained in the NRT aerosol processing.

**5. NRT and SA product comparisons**

**5.1. Total AOD**

In this section, geographic distributions of MISR AOD retrievals from SA and NRT products are analyzed. The datasets encompass three months, March, April, and May of 2020. The NRT retrievals are screened with the revised ARCI threshold of 0.18 as suggested in section 4.3.4. The spatial overlap of the SA and NRT data is achieved using an intersect of the X_Dim and Y_Dim fields in the two data products.

Figure 7 shows the global distributions of geometric mean AOD from the (a) SA and (b)
NRT products. The retrievals are gridded at 2-by-2-degree spatial resolution. Fig. 7c shows the
AOD difference between the two products (NRT – SA).
The largest AOD differences are seen in areas with climatologically high cloud cover,
especially over the Southern Ocean, and over land in areas where potential snow cover could
be an issue. Over the Southern Ocean the SA AODs are predominantly higher than the NRT
AODs. This is due to the increased ARCI threshold in NRT (0.18 vs. 0.15 in SA) which brings in
more aggressive screening of cloud-contaminated retrievals (Witek et al., 2018b). Over land,
where the ARCI parameter is not available, the gridded NRT AODs tend to be higher than the
SA AODs, which is in part related to the differences in snow/ice mask between the two
products. Still, the AOD differences in Fig. 7c are rather small and reflect sampling issues rather
than any systematic deficiencies in NRT processing. At the same time the lack of cloud
classifiers in NRT does not adversely affect AOD distributions, which is consistent with the
statistical analysis presented in section 4.2.3.

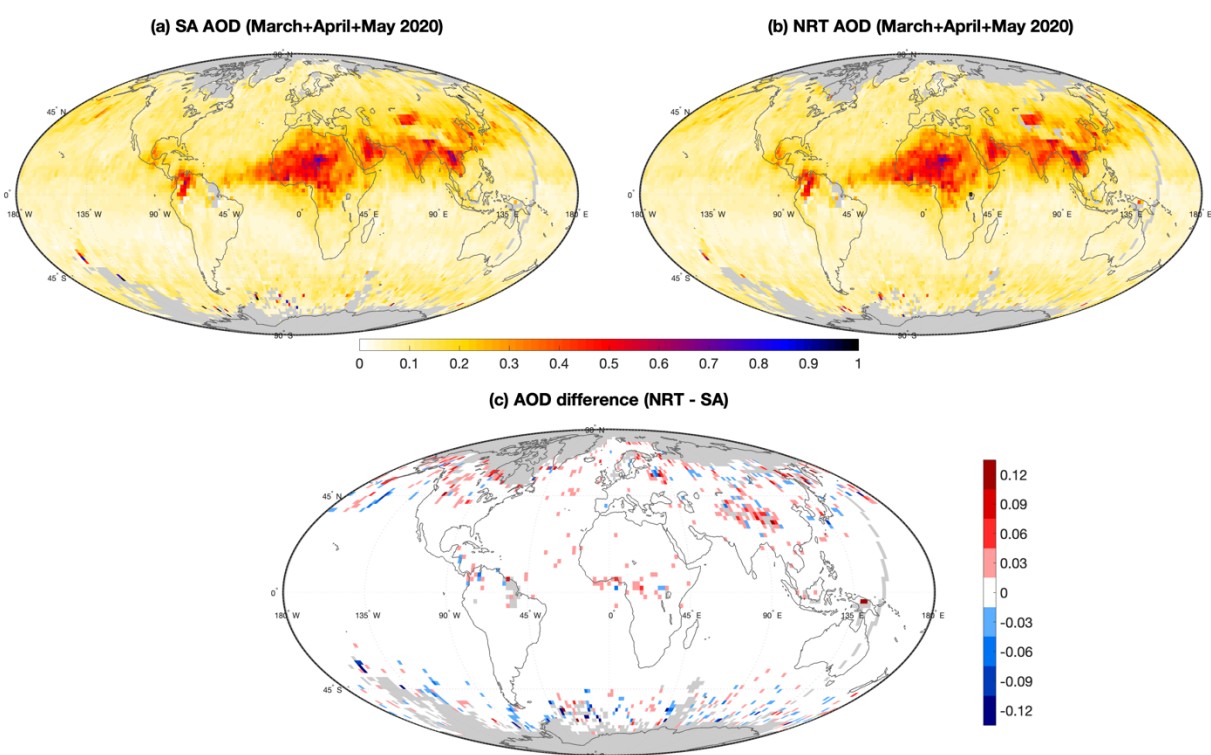


*Figure 7 (a) Global distribution of SA AOD geometric mean values across March, April, and May of 2020 on a 2-by-2-degree*
*spatial resolution; (b) same as in (a) but for NRT AOD; and (c) AOD difference between SA and NRT. Grid points with less than 15*
*retrievals are excluded.*
**5.2. Retrieval yields**
Figure 8 complements Fig. 7 by showing (a) the SA retrieval count distribution as well as (b) the
retrieval count difference between the SA and NRT products.

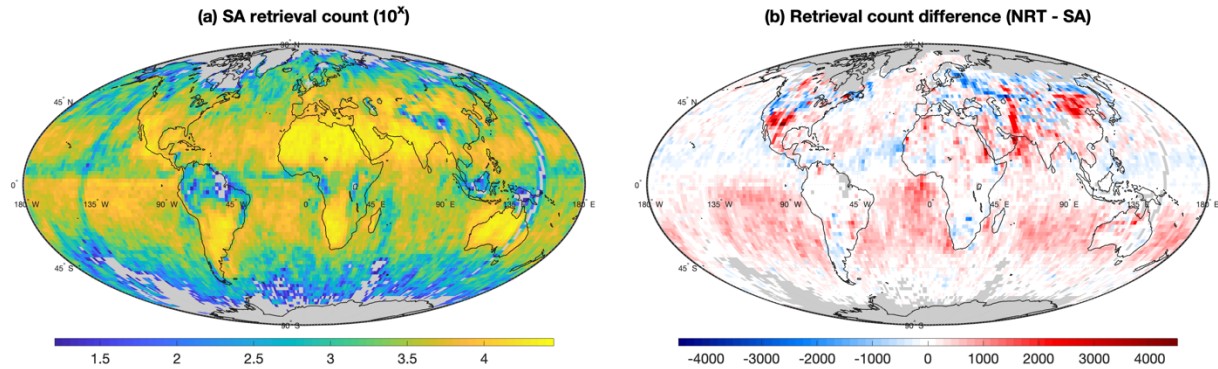

*Figure 8 (a) Decimal logarithm of the retrieval count from the SA product in March, April, and May of 2020; (b) retrieval count*
*difference between SA and NRT. Presented values are gridded at 2-by-2-degree spatial resolution and grid points with less than*
*15 retrievals are excluded.*
The highest number of retrievals is found over the subtropical continents where the
cloud cover is usually the smallest. Over the subtropical oceans in the Southern Hemisphere the
NRT retrieval counts are typically higher than in SA, which results from the absence of upstream
cloud classifiers in NRT processing and subsequently fewer subregions being excluded as
cloudy. Note that this increase in retrieval count caused by the lack of cloud classifiers is not
compensated by the increased ARCI threshold in NRT processing (ARCI$\geq$0.18), which always
reduces the number of retrievals when compared to the default SA threshold (ARCI$\geq$0.15). The
lack of hemispheric symmetry in this case is likely due to the seasonal variability (only months in
northern spring are analyzed here). Over land the lack of upstream cloud classifiers also results
in higher number of NRT retrievals in certain regions, but the surface type exclusion rules
reverse this pattern, especially at higher latitudes. The conservative cloud logic over snow/ice
surfaces in NRT processing often results in the lower number of NRT retrievals in the high
latitudes of the northern hemisphere.
A metric relevant to the potential use of the NRT product in data assimilation is the
retrieval yield per model grid point. The retrieval yield can be measured as, for example, the
number of 1º x 1º grid cells that have at least 15 valid satellite retrievals in them. From this
perspective, the NRT product has a retrieval yield that is about 0.7% higher than the SA
product, based on the three months of data analyzed in this study.

**5.3. Fractional AOD**

MISR's multi-angle retrieval approach enables characterization of aerosol optical and
microphysical properties, such as fractional AODs associated with particle absorption,
nonsphericity, and size (see e.g., Kahn and Gaitley, 2015). This attribute of the MISR SA
product has been applied to many climate and air quality studies and inclusion of this capability
in the NRT product would benefit data assimilation for numerical prediction of atmospheric
aerosols (Benedetti et al., 2018). Consequently, this section provides preliminary statistical
comparisons of the SA and NRT absorption AOD along with small-mode, large-mode, and
nonspherical AOD. The results shown in Fig. 9 indicate that the probability density functions of
these aerosol properties in the NRT product are statistically equivalent to the SA product. This
assessment reaffirms the consistency of the NRT and SA products. Future studies will examine
geographic and statistical differences and other particle properties in more detail.

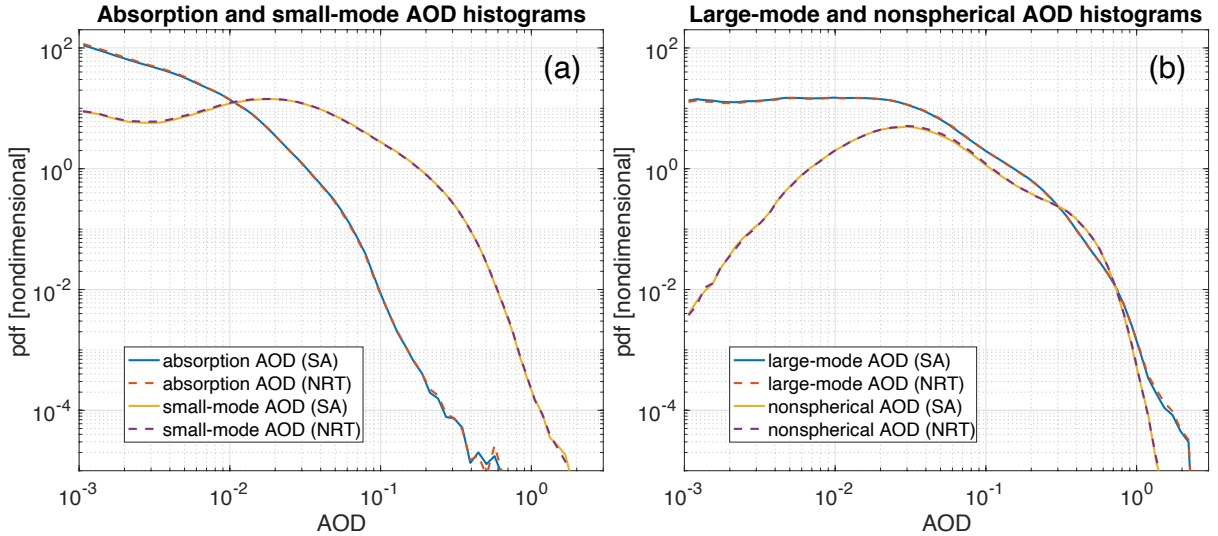


*Figure 9 Normalized probability density functions for select MISR particle property retrievals in March, April, and May 2020.*
*Solid lines represent SA retrievals and dashed represent NRT retrievals. (a) absorption AOD and small-mode AOD retrievals; (b)*
*large-mode AOD and nonspherical AOD retrievals. The differences between the SA and NRT products are negligible.*

**6. Summary**

The MISR V23 aerosol product, publicly available since mid-2018, is a high-resolution state-of-
the-art data product from NASA's Terra flagship mission. V23 AOD retrievals have remarkable
accuracy compared against ground-based observations (Garay et al., 2020; Tao et al., 2020;
Witek et al., 2019) and the product is more intuitive and easier to use than previous versions.
The product is available within 2 days from satellite overpass as a FIRSTLOOK version, and
within 3-to-6 months as a final science-quality SA version that employs the most up-to-date
ancillary datasets. In response to the needs of operational user communities, a new MISR L2
NRT aerosol product has been developed with a 3-hour latency.

The new NRT algorithm does not depend on the upstream cloud classifiers that are

generated in L1 and L2 cloud processing. The lack of cloud classifiers is in large part mitigated
by the aerosol algorithm's built-in cloud identification methods. Analysis of the prototype NRT
product has shown an increased frequency of high-AOD retrievals, especially over oceans and
in climatologically cloudy areas, likely due to an increase in cloud contamination. Adjusting the
ARCI threshold in DW retrievals proves highly effective at eliminating some of these high-AOD
outliers and improves the NRT product's statistical agreement with the SA version. The new
NRT aerosol product applies an ARCI threshold of 0.18 to mitigate cloud contamination in the
absence of upstream cloud masks in NRT processing. The remaining differences in statistical
and geographic distributions between the NRT and SA AODs, which includes information from
the L2 cloud product, are small and largely confined to areas with high cloud cover.

The results of this study also serve as an example of the effects of screening threshold

adjustments in MISR aerosol retrievals on AOD statistics and distributions. Researchers
interested in particular applications and/or specific geographic regions are encouraged to
experiment with their own threshold to achieve most optimal results. The NRT aerosol product
contains both the recommended product contained within the main science directory
"4.4_KM_PRODUCTS" that has the stricter ARCI threshold (ARCI$\geq$0.18), and the unscreened
product without the additional cloud and ARCI filtering designed for more experienced users,
located within the AUXILIARY group.

**Acknowledgements**
This research was carried out at the Jet Propulsion Laboratory, California Institute of
Technology, under a contract with the National Aeronautics and Space Administration. Support
from the MISR project is acknowledged. Special thanks to Andrew Sayer, Jeffrey Reid, and one
anonymous reviewer for carefully reading the manuscript and providing valuable comments. We
would also like to thank Ralph Kahn for providing feedback on the manuscript.

**Data availability**
The MISR V23 SA and NRT data is publicly available and can be downloaded from
https://asdc.larc.nasa.gov/project/MISR. MISR NRT data is not stored permanently and is only
available for three to six months from the time of acquisition; please contact the corresponding
author to request the NRT data from the months analyzed in this study.

**Author contributions**
MLW conceptualized the study, performed the analyses, and prepared the manuscript. MAB
processed the initial NRT data and provided technical support. All coauthors assisted with the
analyses and provided feedback on the results. Furthermore, AMN, FCS, and DJD contributed
to the writing and editing of the manuscript.

**Competing interests**
The authors declare that they have no conflict of interest.

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
