# Peer review of "1. Introduction"

_Atmospheric Measurement Techniques, 2021_

## Referee Comment (RC2)

[referee-annotated manuscript omitted]

---

## Author Comment (AC1)

Authors response to referee Andrew Sayer
Manuscript number: amt-2021-71
"Introducing the MISR Level 2 Near Real-Time Aerosol Product"

**General comments and recommendation:**

This paper describes a near real time (NRT) version of the standard MISR aerosol product. NRT data are particularly useful for assimilation and monitoring purposes, so this development is welcome as the standard aerosol product has latency of several months. There is also a FIRSTLOOK product produced with latency of several days. The main differences between the NRT and standard algorithms are that the former uses different ancillary data (also the case for FIRSTLOOK) and cloud masking because those required for standard product are not available sufficiently fast. Additionally, NRT files are split into partial orbits (sessions).

The work is in scope for the journal. The quality of writing and presentation is high. My main issue with the analysis is that the focus of the comparison is between the FIRSTLOOK and NRT products, and not the "final" standard product and NRT. This is relevant because most climatological and validation analyses use the final product and not FIRSTLOOK. I understand that FIRSTLOOK has been the lower-latency alternative to the standard final product, but think it is worth extending the analysis to also show comparisons to the final standard product. This is because differences in ancillary data (e.g. wind speed over water) could lead to regional systematic differences in retrievals; even if not it would be good to quantify the sensitivity of the algorithm as a measure of retrieval "noise" added by the ancillary data treatment. So the present comparison is mostly examining the effects of the pixel selection criteria, and missing the effects from these other sources. The paper is not too long so I hope the authors would consider adding this in to some parts of the study. Note I am also not aware of an analysis of the differences between FIRSTLOOK and final AOD from the standard product, so adding the standard here will also be informative of that.

Re: We fully agree with the reviewer regarding this issue. When we started working on NRT analysis and on writing of this paper the standard aerosol (SA) product was not yet available, which motivated the use of the FIRSTLOOK product. However, right now SA is available. We therefore repeated our investigations using the SA product. All figures and analyses have been modified accordingly. Furthermore, we included both the FIRSTLOOK and SA results in Figure 3, which shows the overall AOD histograms for land and DW retrievals. The histograms of FIRSTLOOK and SA AODs are almost indistinguishable from each other, which confirms strong similarity of the two products. This is expected, as the ancillary datasets that feed into these two retrievals do not change considerably. Note that the wind speed data in MISR retrievals has three coarse bins centered at 2, 5, and 7.5 m/s. Furthermore, the algorithm attempts wind speed retrieval using glint pattern whenever conditions are favorable, which further limits the

impact of wind speed database on AOD retrievals. None of the conclusions that we reached by using FIRSTLOOK had to be adjusted after we switched to the final SA product.

Other than that, I recommend publication following minor revisions.

**Specific comments:**

Lines 115-117: the authors mention the pixel-level AOD uncertainties. It would be good to make a statement about how these compare between NRT and standard products. It might be that they are near identical (in which case probably only a sentence is needed), while it might be that they show some differences. This could be important because NRT applications (e.g. data assimilation) often need an error model.

Re: We looked at the histograms of AOD uncertainties and found that NRT and standard aerosol (SA) products are in very close agreement, as shown in the figure below. The geographic distributions are also very similar, with small differences aligning with the areas where the differences in coverage are noticeable. We modified Figure 2 by adding a second panel showing the histograms of UNC retrievals in SA, $NRT_{prot}$, and $NRT_{prot}$ gained.

[Figure]

Line 137: the paper says that the NRT aerosol products "are available" (present tense) at https://asdc.larc.nasa.gov/project/MISR . I checked and the only level 2 NRT products listed at the time of posting this review (April 2 2021) are two versions of cloud motion vectors. If the NRT aerosol product is not available at the time the paper is accepted, then the language in the paper should be changed.

Re: Unfortunately, operational processing of the NRT aerosol product has been delayed by a few months. We anticipate that the product will become available in May 2021 before the description paper is officially published.

Line 277: the authors introduce NRT_prot here but this is not used in figures – I think it would be clearer to use this explicitly where NRT_prot is being used (which I think is most analysis before Section 5?). Or am I misunderstanding this?
Re: Good point, it obviously should have been $NRT_{prot}$ instead of NRT. This has been corrected.

Figures 2, 3: legend says "FIRTLOOK" rather than "FIRSTLOOK". I do not think Figure 2 is necessary anyway, since Figure 3 contains the same information in a more useful way (land vs. dark water split).
Re: "FIRSTLOOK" was substituted with "SA" (standard algorithm). We still think showing the histogram in Figure 2 might be useful for some readers as a quick look at the overall performance of the $NRT_{prot}$ algorithm. We furthermore added a second panel in Fig. 2 to show the histograms of retrieved AOD uncertainties.

Line 301: I am not sure I agree with this statement. Looking at Figure 3b, it looks like the NRT_gained pdf over land is flatted overall (more low and more high values than FIRSTLOOK).
Re: We softened the language here to read: "including a slightly flattened but still relatively comparable distribution of the "$NRT_{prot}$ gained" retrievals."

Line 304: I think this citation should be to Sayer and Knobelspiesse (2019) rather than Sayer et al (2020): https://acp.copernicus.org/articles/19/15023/2019/
Re: Yes, the citation has been corrected.

Table 1: is there an error in the count row, or am I misunderstanding? For example, if FL is 50.4 and NRT is 53.7 then shouldn't NRT_gained be 3.3 and not 5.4? The same comment applies to the other elements of this row.
Re: The NRT and SA datasets are compared using the same range of MISR blocks within each orbit. This is because session-based NRT files provide a range of MISR blocks (1 to 180) that they cover, which in turn allows us to select the same range of blocks from the SA product. However, the retrievals are not correlated one-to-one, meaning that there might be NRT retrievals within a block that do not have a SA equivalent, but also that there might be SA retrievals within a block that do not have a NRT equivalent. For that reason, the number of $NRT_{prot}$_gained is not a simple subtraction of the number of $NRT_{prot}$ and the number of SA retrievals. We added the following clarification in the text:
*"Note that the number of $NRT_{prot}$ gained is not the same as the number of $NRT_{prot}$ minus SA. This is because there are cases when a SA retrieval does not have its $NRT_{prot}$ equivalent, making the SA count larger than it would have been otherwise."*

Tables 1, 2, 3: I wonder if it would also be useful to add rows indicating (arithmetic or geometric) AOD standard deviation for each case here. The shift (or lack of) in the mean is one thing but the truncation in variability is another. For example as one shifts to tighter cloud thresholds, Tables 2 and 3 show that the mean AOD does not change too much, but from the corresponding Figures it looks like there is some loss at both the low and high ends (possibly cloud shadows and clouds?). Adding standard deviation would be a quantification of how much this narrows the distribution (might also be negligible, it'shard for me to guess looking at the figures).

Re: We added arithmetic standard deviations to their respective means in Table 2 and 3 (but not in Table 1) as recommended. The standard deviations do not vary much when we change CSP and CSP9 thresholds, similarly to the mean and geometric mean values. When we change the ARCI threshold the standard deviations decrease gradually, getting closer to the standard deviation for the standard aerosol (SA) product. These STD decreases are again in overall agreement with the decreases we see in the mean and geometric mean values.

Line 322: I think "less" should be "fewer" as in principle this is a countable quantity.
Re: Corrected. Thanks for pointing it out.

Figure 7c: the colour scale makes it a bit difficult to see the absolute values here. Most of the world appears white but it's not clear whether this means +/-0.01, +/- 0.02 or what. I wonder if truncating the axis range (somewhere between 0.1-0.15 would probably work) and/or using a discrete or different colour table would make this clearer? Also, at present, the south polar region is in white ("zero difference") but I think should be grey ("no data").
Re: We changed the color scale in the figure. The updated plot has a reduced range of values (from -0.135 to 0.135) and 9 discrete colors with intervals of 0.03.

Figures 7, 8: I wonder if these should be presented as "NRT-FIRSTLOOK" (and hopefully "NRT-standard", see general comments) instead of "FIRSTLOOK-NRT"? That would seem more natural to me as we are comparing what is new (NRT) relative to the previous baseline (FIRSTLOOK or standard). It is possible that this is in fact what is done and the current headers are a typo. For example line 446 says NRT counts are lower than FIRSTLOOK in the southern subtropical oceans, but Figure 8b (if the plot title is correct) shows the opposite.
Re: Yes, we change the difference plots to NRT-SA. There was also a mistake in the text as the subtropical oceans have more NRT retrievals than SA (and FIRSTLOOK for that matter). This is due to the absence of upstream cloud classifiers in NRT processing and subsequently fewer subregions being excluded as cloudy.

Lines 469, 473-476: here the authors make a statement about what ARCI screening threshold is applied in the NRT product, and then make a suggestion that data users do their own experimentation for their particular use case. It is not clear to me from reading whether the

NRT product contains an "unscreened" data set plus ARCI values so that users can do this (as the standard product does)? Or does it only contain retrievals prescreened with the ARCI 0.18 threshold? Somewhere in the paper it would also be good to discuss NRT file contents (whether the file spec is the same as FIRSTLOOK/final or not – if so only one sentence is needed).

Re: We added the following clarification at the end this paragraph:

*The NRT aerosol product contains both the recommended product contained within the main science directory "4.4_KM_PRODUCTS" that has the stricter ARCI threshold (ARCI$\geq$0.18), and the unscreened product without the additional cloud and ARCI filtering designed for more experienced users, located within the AUXILIARY group.*

---

## Author Comment (AC2)

Authors response to anonymous referee #2

"Introducing the MISR Level 2 Near Real-Time Aerosol Product",
by Marcin L. Witek et al., Atmos. Meas. Tech. Discuss.,
https://doi.org/10.5194/amt-2021-71-RC2, 2021

The Standard MISR aerosol optical depth data product is a well established data set which has been used for multiple reanalyses and investigations related to aerosol forcing of climate and air quality research. The recently availability of the NRT MISR aerosol product has opened the door for time sensitive applications such as air-quality forecasting and monitoring. Therefore, a peer reviewed publication documenting the strengths and weaknesses of the algorithm is not only welcome but necessary for proper use of the data. Therefore, I strongly support its publication.

The paper is well written and for the most part documents the shortcoming of the NRT product, primary due to the absence of upstream datasets that impact the cloud screening process. Currently, there are 3 MISR aerosol products: 1) the NRT product being documented here, 2) a FIRSTLOOK product that has similarities to the NRT product, and 3) the final, refined Standard product that is available with 3-6 months latency. While the MISR Final Product in 3) is the golden standard among all their products, it is curious that the authors chose the FIRSTLOOK product, which suffers from many of the same limitations, as a reference. I strongly encourage the team to redo the calculations using the MISR Standard product as reference.

Please see in-line comments in the attached document for additional suggestions for improving the manuscript.

Re: We'd like to thank the referee for reading our manuscript and providing very useful comments and suggestions. We followed the general remark regarding redoing our analysis using the MISR Standard Aerosol (SA) product rather than the FIRSTLOOK product. When we started working on NRT analysis and the writing of this paper, the standard aerosol (SA) product was not yet available, which motivated the use of the FIRSTLOOK product. Since the SA product has become available by now, we repeated our investigations using the SA product instead of FIRSTLOOK. All figures and analyses have been updated accordingly. Furthermore, we included both the FIRSTLOOK and SA results in Figure 3 (see below as well), which shows the overall AOD histograms for land and dark water (DW) retrievals. The histograms of FIRSTLOOK and SA AODs are almost indistinguishable from each other, which confirms strong similarity of the two products. This is expected, as the ancillary datasets underlying the two retrievals do not change considerably. Note that none of the conclusions in the original manuscript, which were based on the FIRSTLOOK product, requires to be adjusted based on the use of the final SA product.

[Figure]

Figure 1 AOD pdfs for land (a) and DW (b) retrievals, respectively. Data statistics are provided in Table 1.

---

## Author Comment (AC5)

[revised manuscript text omitted]

Formatted … [2]

Deleted: 5… more retrievals than SA *FIRSTLOOK* (see Table 1). 5.54…million NRTprot retrievals do not have a matching SA*FIRSTLOOK* …retrieval (NRT gained), and the majority of them (678…) are DW retrievals. The overall geometric means are almost identical in SA FIRSTLOOK … [3]

[Figure]

| | | | | | | | | |
|---|---|---|---|---|---|---|---|---|
| *mean* | 0.168 | 0.169 | 0.171 | 0.111 | 0.115 | 0.146 | 0.240 | 0.243 |
| *geomean* | 0.111 | 0.112 | 0.122 | 0.083 | 0.085 | 0.106 | 0.160 | 0.162 |

*Table 1 Additional statistics for the data presented in Figs. 2 and 3 (statistic for FIRSTLOOK not shown). NRT gained stands for*
*the prototype NRT retrievals that do not have a matching SA equivalent; geomean stands for the geometric mean AOD.*

**4.3.2. Sensitivity to CSP and CSP9 thresholds in DW retrievals**

One way to screen potentially cloud-contaminated high-AOD retrievals is to adjust thresholds on
CSP and CSP9 parameters (Garay et al., 2020). This is furthermore justified by the fact that in
the absence of RCCM, SDCM, and ASCM in NRT$_{prot}$ processing, fewer cloudy subregions are
identified in a retrieval area and consequently CSP and CSP9 have by default lower values.
This argument provides strong justification for investigating sensitivity to increased CSP and
CSP9 thresholds in the NRT$_{prot}$ processing.
The SA product uses the thresholds of CSP=0.7 and CSP9=0.5 (Garay et al., 2020);
when the values of CSP and CSP9 are below these thresholds in a retrieval region, the aerosol
retrieval is removed from the data field recommended for users. Figure 4 and Table 2 show *pdf*s
and AOD statistics for different thresholds of CSP and CSP9 parameters in the NRT$_{prot}$ product
over dark water surfaces. There are only minor changes in the *pdf*s when the thresholds are
increased, including in the high-AOD regime. The mean and geometric mean decrease
gradually but slowly; even at the highest considered thresholds (0.85 for CSP and 0.75 for
CSP9) these statistics are still above the SA values. At the same time the number of passing
NRT$_{prot}$ retrievals decreases considerably faster, with almost 19% of retrievals lost when the
highest thresholds are used. These results indicate that adjusting CSP and CSP9 thresholds is
not an effective strategy to constraining NRT$_{prot}$ retrievals.

[Figure]

**DW NRT retrievals (March+April+May)**

NRT$_{prot}$ with:
— CSP≥0.7
— CSP≥0.73
— CSP≥0.76
— CSP≥0.79
— CSP≥0.82
— CSP≥0.85

*Figure 4 Prototype NRT AOD pdfs over dark water surfaces from spring 2020 obtained with different CSP and CSP9 cloud-screening thresholds. Data statistics are provided in Table 2.*

| N (×10^6) | 30.7 | 30.1 (-1.9%) | 28.4 (-7.4%) | 27.7 (-9.8%) | 25.9 (-15.6%) | 24.9 (-18.9%) | SA 27.6 |
|---|---|---|---|---|---|---|---|
| CSP | ≥0.7 | ≥0.73 | ≥0.76 | ≥0.79 | ≥0.82 | ≥0.85 | |
| CSP9 | ≥0.5 | ≥0.55 | ≥0.6 | ≥0.65 | ≥0.7 | ≥0.75 | |
| mean | 0.1151 ± 0.1200 | 0.1149 ± 0.1199 | 0.1145 ± 0.1190 | 0.1144 ± 0.1191 | 0.1142 ± 0.1185 | 0.1143 ± 0.1189 | 0.1110 ± 0.1079 |
| geomean | 0.0850 | 0.0847 | 0.0841 | 0.0839 | 0.0834 | 0.0832 | 0.0826 |

*Table 2 Additional statistics for the data presented in Fig. 4. Values for CSP and CSP9 indicate their corresponding thresholds for screening AOD retrievals. The arithmetic mean values are accompanied by their respective ± one standard deviations.*

**4.3.3. Sensitivity to ARCI threshold in DW retrievals**

[Figure]

Formatted ... [6]

V23 of the MISR aerosol product introduced a new parameter, called the aerosol retrieval
confidence index (ARCI), that is used to screen high-AOD retrieval outliers caused by cloud
contamination and other factors (Witek et al., 2018b). ARCI, defined only for DW retrievals,
proved to be an efficient metric at filtering out potentially cloud-contaminated AOD retrievals. In
standard processing, retrievals with ARCI < 0.15 are removed from the recommended user
field, but are retained in the AUXILIARY group. The 0.15 threshold is well supported through
statistical analysis (Witek et al., 2018b), although some erroneous AODs still pass this
screening method, suggesting that increasing this threshold might be beneficial in NRT
processing.
Figure 5 and Table 3 show *pdf*s and AOD statistics for different thresholds of ARCI in the
NRT$_{prot}$ product. In this case the differences between ARCI thresholds are quite noticeable,
especially in the high-AOD range of retrievals. Increasing the ARCI threshold to 0.2 leads to a
loss of about 11% of NRT$_{prot}$ DW retrievals, but the resulting mean and geometric mean are
lower than the SA values. At the same time, the absolute number of NRT$_{prot}$ DW retrievals (27.4
million) is still comparable to the number of SA DW retrievals (27.6 million). The *pdf*s and the
statistics suggest that increasing the NRT$_{prot}$ ARCI threshold from 0.15 to 0.18 leads to a
product that has similar characteristics to SA.

| Deleted: FIRSTLOOK |
| Deleted: 8 |
| Deleted: *FIRSTLOOK* |
| Deleted: 9 |
| Deleted: *FIRSTLOOK* |

**DW retrievals (March+April+May)**

[Figure]

NRT$_{prot}$ with:
- ARCI≥0.15
- ARCI≥0.16
- ARCI≥0.17
- ARCI≥0.18
- ARCI≥0.19
- ARCI≥0.20
- SA

[revised manuscript text omitted]

Subscript

| Page 11: [2] Formatted | Microsoft Office User | 4/19/21 4:51:00 PM |
|---|---|---|

Subscript

| Page 11: [2] Formatted | Microsoft Office User | 4/19/21 4:51:00 PM |
|---|---|---|

Subscript

| Page 11: [3] Deleted | Microsoft Office User | 4/19/21 4:00:00 PM |
|---|---|---|

| Page 11: [3] Deleted | Microsoft Office User | 4/19/21 4:00:00 PM |
|---|---|---|

| Page 11: [3] Deleted | Microsoft Office User | 4/19/21 4:00:00 PM |
|---|---|---|

| Page 11: [3] Deleted | Microsoft Office User | 4/19/21 4:00:00 PM |
|---|---|---|

| Page 11: [3] Deleted | Microsoft Office User | 4/19/21 4:00:00 PM |
|---|---|---|

| Page 11: [3] Deleted | Microsoft Office User | 4/19/21 4:00:00 PM |
|---|---|---|

| Page 11: [4] Deleted | Microsoft Office User | 4/22/21 3:30:00 PM |
|---|---|---|

| Page 11: [4] Deleted | Microsoft Office User | 4/22/21 3:30:00 PM |
|---|---|---|

| Page 11: [5] Deleted | Microsoft Office User | 4/19/21 12:18:00 PM |
|---|---|---|

| Page 11: [5] Deleted | Microsoft Office User | 4/19/21 12:18:00 PM |
|---|---|---|

| Page 13: [6] Formatted | Microsoft Office User | 4/22/21 10:13:00 AM |
|---|---|---|

Font: 9 pt

| Page 13: [6] Formatted | Microsoft Office User | 4/22/21 10:13:00 AM |
|---|---|---|

Font: 9 pt

| Page 15: [7] Deleted | Microsoft Office User | 4/19/21 12:42:00 PM |
|---|---|---|

| Page 15: [7] Deleted | Microsoft Office User | 4/19/21 12:42:00 PM |
|---|---|---|

---

## Author Response (AR2)

This paper provides a description of a new NRT MISR product being generated at LANCE for intended use in the operations and hazards community. Most of the discussion is rightfully focused on modifications of the product to account for missing upstream inputs to the operational product, most notably for cloud and snow/ice screening. Their conclusion is that for the most part the NRT product is very similar to the operational product, with vernally increased retrieval yield and occasional cloud screening differences. I am sympathetic to and even appreciative of the author's efforts to create this product. MISR has long suffered from its relatively slow processing, and I am sure the team hopes this will enable broader use by the community. Overall, I think this is a paper worthy of publication in that, for the topics they cover, what is there to argue with? They documented their product differences from the operational version quite well and thus, in some ways, it is good to go. However, I would like to point out that the authors do not address many of the topics that the operation community is concerned with. In this regard, I point the authors to "Benedetti, et al., Status and future of numerical atmospheric aerosol prediction with a focus on data requirements, Atmos. Chem. Phys., 18, 10615–10643, https://doi.org/10.5194/acp-18-10615-2018, 2018." Addressing the concerns outlined in Benedetti et al in this paper would improve the value of this paper and will gain the respect of the user community.

Benedetti et al., points specifically towards the necessary principles of availability, low latency, and product characterization required in order to get the attention of the operational community. Availability? In LANCE, check! Low latency? 3 hour delivery, check! Characterization? Hmmmm… In regard to the NRT product baselining against the standard operational product for coverage and cloud screening, they have largely met the requirement. However, whether the NRT product or even the standard product is worthy of assimilation, as is its professed rationale, that is unclear. Operational developers spend more time cleaning up the satellite products than they do working on the assimilation systems themselves. Three months of data (Mar-Apr-May) of 2020 seems a very short evaluation period with a swath as narrow as MISR's, and does not even account for seasonality. The paper leaves further evaluations to the user community, which quite frankly they are unlikely to do. But again, the "out" for the authors is they can say that they are simply baselining to the standard product, with no other guarantees on data quality.

Figures such as #7, when you see speckled differences between products, suggest the evaluation is under sampled. It would also help if there were more images cases such as in figure 6 where the authors could dive deeper into differences. Indeed, cloud masking is probably the user community's number one concern. It is my recollection that v23 did have much tighter cloud screening and improved quality flags, but the retrieval yield for the highest confidence flag was pretty low. I very much would like to see more discussion on this, e.g. what is the probability of, say 15 samples being collected in, for example, a 1 degree box. Even picking a single day, or two or three consecutive days at random and mapping the values and the differences between the products will help the user understand what is going on in terms of yield, differences in products and residual cloud and/or ice masking errors. Indeed, from a NRT point of view, seasonal averages mean little. It is individual passes and retrievals that need to be evaluated. I also would suggest a little more discussion on what MISR brings to the table relative to other NRT products. For the MISR instrument, you may want to point out where MISR has historically been assimilated in reanalyses. The results are a bit mixed as, on one hand, by its

nature MISR is in the glint region of MODIS. So over water, with MISR+MODIS together, you do get a more full frame during Terra overpasses. But statistically, this does not manifest all that well in RMSE's because of the narrow swath relative to MODIS which swamps the overall signal (Zhang et al., Evaluating the impact of multisensor data assimilation on a global aerosol particle transport model, J. Geophys. Res. Atmos., 119, 4674– 4689, doi:10.1002/2013JD020975, 2014). Playing the game of "adding more AOD data points" to a crowded MODIS, VIIRS, and now geostationary field is unlikely to score MISR big points. But, MISR is superior in products like fine mode fraction, and its ability to semi quantitatively isolate absorption, and identify non-spherical particles has always been MISR's more unique capability. However, these products are not even discussed in this manuscript. I am very curious, does the NRT cloud screening change fine mode fraction or non-sphericity over ocean? Any changes in absorption in biomass burning areas?

Lastly, there are a few things that they could probably do better, although in the context of this paper it is sort of moot. For one, it does not make much sense to use prior year wind speed (I assume this is a monthly average from TASC). This will lead to year by year variability in the product. If you wish to use a climatology, that is fine, but use a static value. For FIRSTLOOK, if the assumption is that the developers want to take into account climate change, they should probably do a 5 year moving window, something that can at least account for ENSO. If operational centers want to make corrections based on their current wind speeds, using the "last year" creates a more difficult moving target. Further, I believe LANCE could use model wind fields. Using a 12 hour wind forecast from GFS would probably help out quite a bit for remote oceans.

Anyways, hope these comments help. My colleagues and I are always grateful when a new NRT product comes into production. Feel free to reach out to me if you wish to discuss further.

Jeffrey S. Reid, US Naval Research Laboratory.

Re: Dr. Jeff Reid, thank you so much for your review, valuable comments, suggestions, and insights. We did our best to include additional analyses in the manuscript based on the data we have available at the moment. Unfortunately, we were not able to extend the evaluation period (March, April, May 2020) with additional months due to the availability of both the NRT and SA datasets. We could include additional months using more recent NRT data, but this would require further delays as the SA product is not yet available for these months. However, per your recommendations, we investigated retrieval yields and particle properties using the available data. This resulted in reorganization of section 5, which now has three subsections:
5.1. Total AOD
5.2. Retrieval yields
5.3. Fractional AOD
We also added a figure showing differences in select particle properties between the SA and NRT products.
For your convenience, we've attached below revised sections 5.2 and 5.3 (new text marked in red). We hope you find our additions valuable.

**5.2. Retrieval yields**

Figure 8 complements Fig. 7 by showing (a) the SA retrieval count distribution as well as (b) the retrieval count difference between the SA and NRT products.

[Figure]

*Figure 8 (a) Decimal logarithm of the retrieval count from the SA product in March, April, and May of 2020; (b) retrieval count difference between SA and NRT. Presented values are gridded at 2-by-2-degree spatial resolution and grid points with less than 15 retrievals are excluded.*

The highest number of retrievals is found over the subtropical continents where the cloud cover is usually the smallest. Over the subtropical oceans in the Southern Hemisphere the NRT retrieval counts are typically higher than in SA, which results from the absence of upstream cloud classifiers in NRT processing and subsequently fewer subregions being excluded as cloudy. Note that this increase in retrieval count caused by the lack of cloud classifiers is not compensated by the increased ARCI threshold in NRT processing (ARCI≥0.18), which always reduces the number of retrievals when compared to the default SA threshold (ARCI≥0.15). The lack of hemispheric symmetry in this case is likely due to the seasonal variability (only months in northern spring are analyzed here). Over land the lack of upstream cloud classifiers also results in higher number of NRT retrievals in certain regions, but the surface type exclusion rules reverse this pattern, especially at higher latitudes. The conservative cloud logic over snow/ice surfaces in NRT processing often results in the lower number of NRT retrievals in the high latitudes of the northern hemisphere.

A metric relevant to the potential use of the NRT product in data assimilation is the retrieval yield per model grid point. The retrieval yield can be measured as, for example, the number of 1º x 1º grid cells that have at least 15 valid satellite retrievals in them. From this perspective, the NRT product has a retrieval yield that is about 0.7% higher than the SA product, based on the three months of data analyzed in this study.

**5.3. Fractional AOD**

MISR's multi-angle retrieval approach enables characterization of aerosol optical and microphysical properties, such as fractional AODs associated with particle absorption, nonsphericity, and size (see e.g., Kahn and Gaitley, 2015). This attribute of the MISR SA product has been applied to many climate and air quality studies and inclusion of this capability in the NRT product would benefit data assimilation for numerical prediction of atmospheric aerosols (Benedetti et al., 2018). Consequently, this section provides preliminary statistical comparisons of the SA and NRT absorption AOD along with small-mode, large-mode, and nonspherical AOD. The results shown in Fig. 9 indicate that the probability density functions of these aerosol properties in the NRT product are statistically equivalent to the SA product. This assessment reaffirms the consistency of the NRT and SA products. Future studies will examine geographic and statistical differences and other particle properties in more detail.

[Figure]

Figure 9 Normalized probability density functions for select MISR particle property retrievals in March, April, and May 2020. Solid lines represent SA retrievals and dashed represent NRT retrievals. (a) absorption AOD and small-mode AOD retrievals; (b) large-mode AOD and nonspherical AOD retrievals. The differences between the SA and NRT products are negligible.

---

## Author Response (AR3)

Title: Introducing the MISR Level 2 Near Real-Time Aerosol Product
Author(s): Marcin L. Witek et al.
MS No.: amt-2021-71
MS type: Research article
Iteration: Minor Revision

Dear Editor,

We are writing to you to kindly ask you to reconsider your recent decision regarding the need for further revisions to our manuscript. The effort involved would be nontrivial. In the review process we addressed in great detail comments and suggestions from the first two reviewers, which resulted in considerable improvements to the manuscript, with the result that these reviewers were fully satisfied with our efforts.

The third reviewer assessed the revised manuscript and while the overall tone of the review was generally positive, he did suggest revisions aimed at addressing specific interests of the data assimilation community. Although we feel that many of the additional analyses requested by the third reviewer exceed the scope of this paper and should instead be addressed in a separate manuscript, we did address his questions regarding differences in retrieval yields and particle properties. Our investigations showed very little difference between the NRT and the standard product with respect to retrieval yields, AODs, and particle properties.

The main goal of the submitted manuscript is to introduce the new NRT aerosol product to the community. We describe important differences and assess performance of the NRT product with respect to the standard product. The primarily statistical approach to our analyses is well aligned with its introductory purpose. When necessary, we show specific examples highlighting certain implications of the NRT processing (Figure 6). However, it is beyond the scope of this paper to analyze in detail specific scenes, orbits, or days.

In light of the fact that the MISR project is releasing the NRT product to the public, we consider timely publication of this introductory paper an important source of information for potential users. As with any satellite remote sensing product, feedback from the community regarding strengths and limitations is essential and the primary goal of this paper is to engage the community in such assessments. Different users will have different perspectives and needs. Consequently, we recommend sharing the revised manuscript with the third reviewer and ascertaining whether the compromise reflected in this revision is acceptable to him so that the paper can move forward.

Kind regards,

  Marcin Witek and co-authors